# Synthesis, Characterization, and Assessment of Anti-Cancer Potential of ZnO Nanoparticles in an In Vitro Model of Breast Cancer

**DOI:** 10.3390/molecules27061827

**Published:** 2022-03-11

**Authors:** Alaa A. A. Aljabali, Mohammad A. Obeid, Hamid A. Bakshi, Walhan Alshaer, Raed M. Ennab, Bahaa Al-Trad, Wesam Al Khateeb, Khalid M. Al-Batayneh, Abdulfattah Al-Kadash, Shrouq Alsotari, Hamdi Nsairat, Murtaza M. Tambuwala

**Affiliations:** 1Department of Pharmaceutics and Pharmaceutical Technology, Faculty of Pharmacy, Yarmouk University, Irbid 21163, Jordan; m.obeid@yu.edu.jo; 2School of Pharmacy and Pharmaceutical Sciences, Ulster University, Coleraine BT52 1SA, County Londonderry, Northern Ireland, UK; bakshi-h@ulster.ac.uk; 3Cell Therapy Center, The University of Jordan, Amman 11942, Jordan; walhan.ctc@gmail.com (W.A.); sh.alsotari@gmail.com (S.A.); 4Department of Basic Medical Sciences, Faculty of Medicine, Yarmouk University, Irbid 21163, Jordan; raed.ennab@yu.edu.jo; 5Department of Biological Sciences, Faculty of Science, Yarmouk University, Irbid 21163, Jordan; bahaa.tr@yu.edu.jo (B.A.-T.); wesamkh@yu.edu.jo (W.A.K.); albatynehk@yu.edu.jo (K.M.A.-B.); 6Faculty of Medicine, The University of Jordan, Amman 11942, Jordan; kadash1@yahoo.com; 7Pharmacological and Diagnostic Research Center, Faculty of Pharmacy, Al-Ahliyya Amman University, Amman 19328, Jordan; hammdi2000@yahoo.com

**Keywords:** zinc oxide nanoparticles, anti-cancer therapy, triple negative, nanomedicine, green synthesis

## Abstract

Advanced innovations for combating variants of aggressive breast cancer and overcoming drug resistance are desired. In cancer treatment, ZnO nanoparticles (NPs) have the capacity to specifically and compellingly activate apoptosis of cancer cells. There is also a pressing need to develop innovative anti-cancer therapeutics, and recent research suggests that ZnO nanoparticles hold great potential. Here, the *in vitro* chemical effectiveness of ZnO NPs has been tested. Zinc oxide (ZnO) nanoparticles were synthesized using *Citrullus colocynthis* (L.) Schrad by green methods approach. The generated ZnO was observed to have a hexagonal wurtzite crystal arrangement. The generated nanomaterials were characterized by transmission electron microscopy (TEM), scanning electron microscopy (SEM), UV-visible spectroscopy. The crystallinity of ZnO was reported to be in the range 50–60 nm. The NPs morphology showed a strong absorbance at 374 nm with an estimated gap band of 3.20 eV to 3.32 eV. Microscopy analysis proved the morphology and distribution of the generated nanoparticles to be around 50 nm, with the elemental studies showing the elemental composition of ZnO and further confirming the purity of ZnO NPs. The cytotoxic effect of ZnO NPs was evaluated against wild-type and doxorubicin-resistant MCF-7 and MDA-MB-231 breast cancer cell lines. The results showed the ability of ZnO NPs to inhibit the prefoliation of MCF-7 and MDA-MB-231 prefoliation through the induction of apoptosis without significant differences in both wild-type and resistance to doxorubicin.

## 1. Introduction

During the last two decades, there has been a surge in interest in producing environmentally friendly metal oxide nanoparticles in materials science and nanotechnology research disciplines. The nanoparticles (NPs) exhibit distinct shape, size, chemical, and physical attributes. Chemical reduction strategies are often utilized on an industrial scale to synthesize large quantities of metal NPs; however, they are more costly, environmentally harmful, and require immense energy. Synthesis of NPs using various noble metals can be used in many biomedical applications, including cancer therapies, selective drug delivery, and molecular imaging [1,2,3,4]. 

The most common cancer in women is breast cancer, with more than 1.7 million confirmed cases diagnosed worldwide. The disease is complicated but is usually identified by the estrogen receptor (ER), the progesterone receptor (PR), and the human epidermal receptor 2 (HER2) [5]. More than 70% of breast cancers have ER expression, 15% overexpress HER-2, and anti-hormonal and anti-HER2 treatments are available in these cases. Untargeted chemotherapy is the only alternative for the remaining patients whose tumors express none of these receptors (the so-called triple-negative breast cancers (TNB) because few known therapeutic agents have yet been developed for this active subtype of breast cancer [6]. Although most patients with breast cancer initially respond well to chemotherapy, a high number of patients return with remote metastases. In general, new customized methods to combat drug resistance must be implemented urgently [7]. In recent times, nanomedicine-mediated approaches have shown a strong interest in cancer treatment due to targeted delivery, high absorption, biocompatibility, and multifunctionality of the generated nanoparticles [8]. Consequently, the need to utilize the ZnO-based system to address the invasive triple-negative breast cancer and address the aggressive phenotype characterized by a significantly high level of metastasis, a poor prognosis, and unpredictability in therapy due to conventional chemotherapy suppressing malignant cells is the main rationale for this work. 

Ionic zinc is an integral component of many regular biological disease cycles, including homeostasis and immune function, oxidative stress, apoptosis, and aging [9]. Zn^2+^ has also impacted cancer and is an appealing chemical treatment, as it mediates gene expression changes, cellular metabolism reduction, and cancer cell apoptosis [10,11,12]. One of the primary benefits of nanomaterials is the potential to manufacture regulated nanomaterials less than 100 nm. The recognizable electronic and optical semiconductor with a wide range of 3.37 eV and high energy (60 meV) at room temperature and a robust direct band difference is zinc oxide (ZnO) between different semiconducting materials [13]. ZnO NPs are more detrimental to breast cancer than healthy cells with high degrees of cancer cell selection [7,14]. Recently, ZnO NPs have been reported to supply therapeutic proportions in contrast to chemotherapy drugs and have a high degree of cancer cell selectivity [15]. Regardless of their method of synthesis, and because of their low toxicity and biodegradability, ZnO NPs have been used in the inhibition of UV radiation [16], antidiabetic activity [17,18], antimicrobial agent [19], anti-inflammatory agent [20], wound healing [21], cancer imaging and diagnosis [22], photocatalysis [16], electronics [23].

Biosynthesis or green synthesis has recently become a means of alternative synthesizing nano-metal oxides [24]. *Citrullus colocynthis* (L.) Schrad has been used as a medicinal plant for the treatment of leprosy, diabetes, cough, common cold, bronchitis, asthma, joint pain, jaundice, toothache, and gastrointestinal diseases, including constipation, indigestion, dysentery, colic pain, and many other microbial infections related to the plant [25,26,27]. It was considered that the repeated use of chemical methods contributes to the appearance on the surface of nanoparticles of certain toxic chemicals that could have a harmful impact in medical applications [28]. Sol-gel processing [29], homogeneous precipitation [30], mechanical milling [31], organometallic synthesis [32], the microwave approach [33], spray pyrolysis [34], thermal evaporation [35], and mechanochemical synthesis [36] are all strategies for producing ZnO NPs. Chemical methodologies and hazardous reducing chemicals, the majority of which are very reactive and harmful to the environment, are often used in these procedures [37]. As a result, biological synthesis approaches were used to synthesize ZnO NPs while minimizing environmental impact. This method has significant benefits, notably environmental friendliness through the limitation of toxic byproducts, biocompatibility, cost-effectiveness, immunogenicity, ease of scaling up, simple laboratory apertures, and the generation of safe-to-use NPs. The plant extract source is expected to induce nanoparticle morphology. Leaf extract contains different concentrations and combinations of organic reducing agents [38,39]. The established hypothesis for the formation of NPs in this technique is a phytochemical-driven response wherein the leaf extracts comprise various reducing chemicals, including enzymes, antioxidants, and phenolic moieties that convert zinc cations into ZnO NPs. The presence of phytochemicals promotes the postulated reduction of Zn(NO_3_)_2_∙6H_2_O to generate zerovalent zinc, which further leads to the agglomeration of zinc atoms into nanosized particles that are ultimately stabilized by phytochemicals to produce isotropic (spherical) ZnO.

Often used as a supplier for chemical therapy products, ZnO NPs have recently been shown to have the ability to combine ZnO NPs with doxorubicin (Dox) to improve Dox-resistant ovarian cancer cells in 3D cancer spheroids, but not of triple-negative Dox-sensitive MDA-M B-231 cells [40]. Of various metal NPs, ZnO NPs demonstrate increased cytotoxicity, resulting in oxidative damage and cell death by producing reactive oxygen species when the cell has exhausted its antioxidant ability [41]. 

ROS have highly activated oxygen metabolites for cellular macromolecules in oxidation, such as lipids, proteins, and polypeptides. In the case of excessive ROS production or decreased intrinsic antioxidant capacity, indiscriminate oxidation induces adverse effects in human ovarian cells resulting from oxidative stress, the equilibrium between ROS production and the antioxidant mechanism is maintained in the cell [42]. Ideally, it would help concentrate on the accelerated separation of cancer cells based on modern cancer therapies. 

A ZnO nanoparticle can quickly absorb UV rays [43]. Through different cellular pathways, zinc is a primary co-factor and has a significant role in sustaining cell homeostasis; thus, ZnO is biocompatible. The ZnO supplied can easily be biodegraded or participate in the active nutritional process of the body. ZnO nanoparticles are intrinsically potent cytotoxic against in vitro cancer cells compared to other nanoparticles [44,45]. Although extracellular ZnO is biocompatible, increased amounts administered intracellular ZnO suggest a rise in cytotoxicity through zinc-mediated protein production mismatch and oxidative stress. Herein, we report a one-step method that uses a greener, environmentally friendly, and safer approach to generate monodisperse ZnO NPs using the leaf extract of *Citrullus colocynthis* (L.) Schrad. We believe such a plant in a harsh environment contains higher quantities of phytochemicals, controlling the generated NPs size and shape. 

## 2. Materials and Methods

### 2.1. Materials

The following chemicals were obtained: zinc nitrate hexahydrate (Zn(NO_3_)_2_·6H_2_O; 98% purity), sodium hydroxide (NaOH; 97% purity), dimethyl sulfoxide (DMSO) of (Sigma-Aldrich, Darmstadt, Germany) analytical grade. An amount of 50 kD MWCO membranes (Spectra/Por7) from REPLIGEN. Corning™ Spin-X™ ultrafiltration devices (0.45 µm). MCF-7 cells (ATCC number: HTB-22) and MDA-MB-231(ATCC number: HTB-26) cells, RPMI-1640 growth medium (Capricorn Scientific GmbH, Hessen, Germany). Eagle’s Minimum Essential Medium (EMEM) (Euroclone SpA, Pavia, Italy). An amount of 10% (*v*/*v*) fetal bovine serum (FBS), L-glutamine, penicillin-streptomycin antibiotics and (3-(4,5-dimethylthiazol-2-yl)-2,5-diphenyltetrazolium bromide) from Thermo Fisher Scientific (Waltham, MA, USA).

### 2.2. NPs Synthesis 

An aqueous solution of 1 M sodium hydroxide solution (NaOH); 4 g in 100 mL of double-distilled (ddH_2_O) water. The resulting NaOH solution was heated to 90 °C on a magnetic hot plate with constant stirring at 500 rpm. Once the solution reached the desired temperature, 0.5 M of an aqueous solution of zinc nitrate hexahydrate (17 g of Zn (NO_3_)_2_∙6H_2_O dissolved in 100 mL of ddH_2_O). The Zn (NO_3_)_2_∙6H_2_O solution was added drop by drop from the sidewalls of the container to the NaOH solution over 40–60 min. The Zn(NO_3_)_2_∙6H_2_O solution to the alkaline solution resulted in the immediate precipitation of ZnO by the color change from transparent to white. Once Zn(NO_3_)_2_∙6H_2_O was added, the solution was stirred for another 2 h on a magnetic hot plate while maintaining the reaction temperature at 90 °C. The solution was centrifuged at 8000× *g* for 30 min to remove any large aggregates before the supernatant was dialyzed for 4 h against ddH_2_O using 50 kDa molecular-weight-cut-off (MWCO) membranes (Spectra/Por7; REPLIGEN). The particles were spun at 10,000× *g* for 30 min; the supernatant was removed; the precipitated particles were air-dried at 50 °C in a dry oven and stored at 4 °C until further use. Once used, the ZnO NPs were suspended at a concentration of 10 mg/mL suspended in ddH_2_O.

### 2.3. Green Synthesis on NPs

Fresh leaves from the Wadi Araba, south of Jordan, were collected from *Citrullus colocynthis* (L.) Schrad, in March 2020. Dr. Wesam Al Khateeb identified the plant sample, and the plant materials were preserved until needed. The frozen leaves (−20 °C) were boiled in the ddH_2_O water for 45 min at 100 °C. The dark brown extract was filtered to remove insoluble fractions and macromolecules. Corning™ Spin-X™ ultrafiltration devices were used (0.4 µm). The supernatant was later stored in brown glass bottles, and the resultant extract was stored in the refrigerator at 4 °C until use. The extract obtained provided polyphenols and amino phytocompounds (protein), which acted as the reducing and capping agent for the generation of ZnO NPs.

### 2.4. Purification of ZnO NPs 

After completion of the reaction, the ZnO NPs were spun at ambient temperature for 15 min at 5000 g (Eppendorf benchtop) to eliminate large aggregates. Supernatants were collected and further purified in the disposable desalting of the PD-10 disposable column (GE Healthcare). The nanoparticles were eluted in a 3.5 mL 10 mM sodium phosphate buffer with a pH of 7.0. ZnO NPs were further purified on dialysis tubes (20 kDa, Spectrum Labs) with buffer exchange every 2 h twice, followed by overnight incubation (15–20 h).

### 2.5. Characterization of Nanoparticles

#### 2.5.1. UV-Vis Absorption Spectra

Spectroscopic analysis was performed to determine the biosynthesis of ZnO NPs. UV-vis was recorded at different time points while the concentration of the plant extract and the aqueous solution of zinc chloride were fixed. Absorption spectra were recorded at ambient temperature in the range of 300 to 800 nm using a Nanodrop 2000 UV-vis spectrophotometer (Wilmington, DE, USA). The absorbance at 374 nm was observed as a characteristic absorption peak for wurtzite hexagonal crystals of ZnO NPs.

#### 2.5.2. Dynamic Light Scattering (DLS)

The mean hydrodynamic diameter of the ZnO nanoparticles was determined by dynamic light scattering (DLS) using a Nano ZS (Malvern Instruments, UK). In addition, the polydispersity index (PDI) has been used as an indicator of the size distribution. Measurement was performed with 20 mW He—Ne laser, λ_0_ = 780 nm, scatter angle = 90 degrees, 1.33 molar refractive index; viscosity at 25 degrees Celsius 0.8872; automatic attenuation was set ranging from 6 to 9.

#### 2.5.3. Zeta Potential (ZP)

The Particle zeta potential was measured on a Zetasizer™ NanoZS-90 (Malvern Instruments) under the settings of 4 mW, λ_632_ nm He–Ne laser with a detector angle of θ = 173° degree. During the measurements, the device’s cell voltage was set to 80 V. The reference beam was intensity 2000 and 3500 kcp. On average, three measurements of each sample and potential zeta values have been reported. All ZnO nanoparticles were diluted in free deionized water containing a final sodium chloride concentration of 1 mM (pH 7.4).

#### 2.5.4. Transmission Electron Microscopy (TEM)

The morphology and size of ZnO nanoparticles were analyzed by TEM. First, copper grids (SPI Supplies, West Chester, PA, USA) were uniformly coated with carbon and glow discharged using a low vacuum Leica EM ACE200 coating system (Leica, Leica, Austria), followed by a 1.5% vinyl K solution chloroform. Next, a drop (10 μL) of water-diluted ZnO nanoparticles (1% *v*/*v*) was placed on a 200 mesh formvar carbon-coated copper grid and left to dry overnight. The prepared grid was then subjected to a TEM examination using Versa 3D (FEI, Eindhoven, NOORD-BRABANT, The Netherlands) operated at an accelerating voltage of 30 kV.

#### 2.5.5. Nanoparticle Tracking Analysis (NTA)

NanoSight LM10 with a laser LM14 module set at a wavelength of 532 nm and NTA 2.3 build 0033 analytical software equipped with a high-sensitivity sCMOS camera were used to measure ZnO NPs hydrodynamic diameter and concentration (particles numbers) utilizing an NPs tracking analysis (Malvern Instruments Ltd., Malvern, UK). Inject suspended samples in ddH_2_O with sterile syringes into the sample chamber until the tip of the solution reached the nozzle was injected into the sample chamber with sterile syringes. The NP size concentration was recorded using ten 30 s videos with a camera level of 7 and a detection threshold of 5 for each video sample. Measurements with 380 gain cameras setting and a shutter speed of 15 ms with auto particle detection adjustments. Before measurements, the instrument was calibrated with 100 nm standard polystyrene particles at a specific concentration.

### 2.6. Cellular Assay

#### 2.6.1. Cells

Cells MCF-7 (ATCC number: HTB-22) and MDA-MB-231(ATCC number: HTB-26) cells were cultured in RPMI-1640 growth medium (Capricorn Scientific GmbH, Ebsdorfergrund, Germany) and Eagle’s Minimum Essential Medium (EMEM) (Euroclone SpA, Via Figino, Italy), respectively. Both RPMI-1640 and EMEM medium were supplemented with 10% (*v*/*v*) fetal bovine serum (FBS), 1% (*v*/*v*) 200 mM L-glutamine, and antibiotics; Penicillin-Streptomycin (100 IU/mL–100 µg/mL). Cells were kept in a humidified 5% CO_2_ at 37 °C. Doxorubicin-resistant MCF-7 (MCF-7/DR) and MDA-MB-231 (MDA-MB-231/DR) cell lines used in this work were previously developed in our lab and maintained using similar conditions of wild-type cell lines [46,47]. 

#### 2.6.2. MTT Viability Assay to Determine the IC_50_ of ZnO NPs

To determine the IC_50_ of zinc oxide for MDA-MB-231_DoxS, MDA-MB-231_DoxR, MCF7_DoxS, and MCF7_DoxR cells. The MTT assay has been performed. Each cell line cell was seeded in a plate of 96 wells plate (4 × 10^3^ cells/well) and incubated for 24 h at 37 °C in a 5% CO_2_ incubator to allow cell attachment. After incubation, the media were replaced with new fresh media treated with serial concentrations of zinc oxide (0 to 100 µg/mL) and incubated for 72 h at 37 °C in a 5% CO_2_ incubator. After incubation, new 100 µL fresh media replaced the old media, and 15 µL MTT (3-(4,5-dimethylthiazol-2-yl)-2,5-diphenyltetrazolium bromide) was added to each well, and the plates were incubated for a further 3 h. Then, 50 µL of DMSO is added for each well to stop the reaction and incubated for 10 min in the darkroom. The absorbance was measured at 570 nm using a Glomax plate reader (Promega, Madison, WI, USA). The IC_50_ values were measured by nonlinear regression analysis using log(inhibitor) vs. variable slope (four parameters).

#### 2.6.3. Apoptosis Assay

To determine the cell death mechanism of breast cancer cell lines and fibroblasts cells treated with ZnO. Annexin V/PI stain was examined by flow cytometry. MDA-MB-231/WT, MDA-MB-231/DR, MCF7/WT, MCF7/DR (2 × 10^5^ cells/well) were seeded in 12-well plates (SPL, South Korea) and incubated at 37 °C for 24 h. Cells were then incubated in triplicate with 1 mL of compatible medium containing 10 µg/mL of ZnO for 24 h. Untreated cells were used as a negative control. Following treatment, cells were harvested using 250 µL of Accutase (Capricorn Scientific, Ebsdorfergrund, Germany) and Eagle’s Minimum Essential Medium (EMEM) (Euroclone SpA, Via Figino, Germany). According to the manufacturer’s instructions, the apoptosis assay was performed using eBioscience™ Annexin V-FITC Apoptosis Detection Kit (Invitrogen, Waltham, MA, USA). Samples were analyzed immediately using FACS Canto II (BD, San Jose, CA, USA).

### 2.7. Statistical Analysis

The statistical analyses were performed using the Student’s *t*-test. The significant difference was considered when the *p*-value was less than 0.05. All values were expressed as mean ± SD.

## 3. Results and Discussion 

### 3.1. Characterization of NPs

#### UV-Vis

Herein, we report a one-step and greener method to produce ZnO NPs. The predicted theory for the generation of NPs in this method is a phytochemical-driven process in which leaf extracts comprise multiple reducing compounds such as antioxidants, enzymes, and phenolic (such as terpenoid, flavonoid, saponin, and phenol) moieties that convert zinc cations into ZnO NPs. Thus, the presence of phytochemicals promotes the hypothesized reduction of zinc cations to create zerovalent zinc, which proceeds to the agglomeration of zinc atoms to nanosized particles, which are eventually stabilized by the polyphenols (quercetin) to generate spherical ZnO NPs. The richly accessible alkaloids and flavonoids were used in plant extract as stabilizers and capping agents [48]. In the current green synthesis protocol, after 4–10 min of incubation of Zn^2+^, the color changed from yellowish-brown to off-white. The reaction mixture of *Citrullus colocynthis* (L.) Schrad leaves extract, and solvent zinc nitrate aqueous leaf extract developed an intense color change. The observed color shift confirmed the formation of ZnO NPs. The absorption spectrum of ZnO reveals a characteristic absorption peak at 374 nm synthesized using zinc nitrate hexahydrate to indicate the formation of ZnO NPs. The reported color variations were attributed to the resonance of ZnO NPs from the surface plasmon. The UV-vis spectra of the biosynthesized zinc oxide nanoparticles with the *Citrullus colocynthis* (L.) Schrad leaves extract display a peak absorption of 374 nm, characteristic of ZnO NPs. In keeping with Mei’s principle, the synthesized nanoparticles’ form is spherical because of a high UV-vis absorption peak [49].

The UV-vis peak variation of the ZnO NPs is due to the difference in size and structure, which results from a variety of salt precursors used in the synthesis and the calcination temperatures. Furthermore, these peaks are due to electronic transitions from the deep process valence range to the conducting band reported in the literature [50]. The observed sharp spectrum peaks measured at ambient temperatures indicate a monodisperse distribution of the generated NPs, and because no other peaks were observed in the spectrum, it confirms that the formulation is of NPs and confirms the wurtzite hexagonal crystalline structure. The maximum absorbance was recorded at 374 nm, as shown in Figure 1. The results obtained agree with the formulation previously reported [51].

Furthermore, the corresponding band gap was calculated using the UV-vis spectrum using the formula E_g_ = 1240/λ with the resultant optical band gap values of the generated ZnO NPs were measured within the range of 3.20 eV to 3.32 eV with the energy gap decreases with the increase in the particles size. Furthermore, the UV-vis spectrum revealed whether nanomaterials produced in this research were pure because the spectrum was smooth beyond the initial decline. Unless the sample had contaminants in the generated nanomaterials, it would have performed as dopants and induced a lower energy transfer (higher wavelength). This will represent minor peaks over the band gap’s original drop, and this finding agrees with what has been published in the literature [52]. The nanoparticles’ monodispersed existence was seen by a significant sharp absorption of ZnO at 374 nm. The efficient mass model measured a particle size diameter and found it 50 nm on average, close to the SEM size. However, the difference in the reported size between DLS and TEM is due to the measurements of the electron-dense part of the NPs. The DLS, in contrast, is related to the particle’s movement in suspension with a different coefficient factor and related to the Stokes–Einstein equation. Thus, the hydrodynamic size represents the NPs transportation abilities and considers any protective (steric layer or surfactant stabilizing) layer surrounding the particle.

The elemental analysis of the generated ZnO NPs using the EDXs technique and the particles of the elemental composition analysis revealed that zinc was 52.68% and oxygen was 47.32% (data not shown). The elemental analysis agrees with the generated ZnO nanoparticles with an almost 1:1 ratio. We believe that the slight variation in this ratio in the NPs composition is due to the loss of the X-rays being detected under the measurement conditions reported previously [53]. 

The hydrodynamic and particle size distribution patterns (DLS) are commonly used to determine the colloidal solution’s particle sizes and polydispersity index. In the current investigation, the DLS analysis showed that the average particle size in the aqueous medium of the prepared ZnO NPs was approximately 50–60 ± 5 nm and a polydispersity index of 0.3 ± 0.03 as shown in Figure 2A, indicating that the generated particles are almost monodispersed and homogeneous. Furthermore, such findings were consistent with the NTA analysis, as shown in Figure 2B. Most nanoparticles showed a monomodal size distribution of 53 ± 3.5 nm and verified their monodispersity with a concentration of 1.43 × 10^11^ nanoparticles mL^−1,^ as shown in Figure 2B, which is consistent with the obtained DLS data. In addition, the NTA video frames display 53 nm in narrow size distribution. NTA is a powerful technique for characterizing DLS and is especially useful for testing the produced nanoparticles’ aggregation and distribution. Repeated NTA measurements (20 experiments) of the generated nanomaterials reported similar size distributions with an average concentration ranging from 1.52 × 10^10^ to 2 × 10^11^ particles/mL. 

Furthermore, the generated ZnO NPs surface charge (ζ) of the suspended colloid with values of −6 ± 1 is summarized in Table 1. At least three zeta potential measurements were determined for each sample, with standard deviation values below two mV for all generated samples. The criterion determines the repulsive force of the interparticle by the charge–charge interaction, representing aggregations or agglomerations of generated particles. All ZnO samples produced showed similar ζ values from different synthesis repeats.

Furthermore, the geometrically rounded highly crystalline NPs generated with some hexagonal edges with an average of ~50 nm corresponding to the hexagonal crystal phase of wurtzite, as shown in Figure 3A. The generated data agree with SEM Figure 3B, which shows round-shaped homogenously distributed NPs on the microscopy grid with occasional overlap showing some agglomerated structures. 

The morphology and crystal structure of the generated particles were obtained with high-resolution TEM images. Figure 4A shows that the ZnO NPs were quasi-spherical in shape. The high resolution of ZnO NPs images depicts the presence of high-quality lattice arrangements without any distortion. The interplanar spacing of adjacent lattice fringes of ZnO NPs was visible and corresponded to the hexagonal wurtzite crystals, and these fringes agreed with the selected area diffraction (SAED). Figure 4B confirms the presence of crystalline nanostructures. Furthermore, the Debye–Scherrer rings were assigned (010), (002), (011), (012), (110), and (103), respectively. 

The MTT assay was performed to explore the cytotoxic effects of the ZnO NPs on MDA MB231/WT, MDA-MB-231/DR, MCF-7/WT, and MCF-7/DR cell lines. These breast cancer cell lines were treated with different ZnO NPs (0 to 100 µg/mL). Figure 5A,B show the dose–response curves and the IC_50_ results of ZnO NPs in each cell line. In the MCF-7 cell line, the IC_50_ results did not show a significant difference between MCF-7/WT cells (1.85 ± 0.4 µg/mL) compared to MCF-7/DR cells (1.5 ± 0.2 µg/mL) (*p* > 0.05) (Figure 5A). On the other hand, in the MDA-MB-231 cell line, the IC_50_s were around two folds higher than the MCF-7 cell line. Furthermore, the results did not show significant differences between MDA-MB-231/WT cells (4.2 ± 0.6 µg/mL) compared to MDA-MB-231/DR cells (4.0 ± 1.1 µg/mL) (*p* > 0.05) (Figure 5B).

Apoptosis assay was performed to investigate the mechanism of cell death results from ZnO NPs treatment. Interestingly, our data showed that both MCF-7/WT and MCF-7/DR cells treated with ZnO NPs revealed a significant decrease in the percentage of viable cells (14% and 34%), compared to untreated control cells of both MCF-7/DR and MCF-7/WT (89% and 75%) (*p* < 0.0001 and <0.001), respectively (Figure 6A,C). Furthermore, our results showed that the late apoptotic percentage increased after ZnO treatment in a statistically significant way. The MCF-7/DR cells showed 41% late apoptotic cell death, whereas MCF-7/WT cells showed 67% late apoptotic cell death than their untreated control cells, 3% (*p* < 0.01) and 11% (*p* < 0.001), respectively. There was inconsistency with the MCF-7 cell line; whereas, MDA-MB-231/DR and MDA-MB-231/WT cells treated with ZnO showed the same death mode (Figure 6B,D) as the number of viable cells decreased significantly for the treated cells of MDA-MB-231/WT (2%) and MDA-MB-231/DR (67%) compared to the untreated control cells (74% and 95%) (*p* < 0.0001 and <0.001), respectively. Similarly, the percentage of late apoptotic cell death was increased when MDA-MB-231/DR and MDA-MB-231/WT were treated with ZnO (89% and 21%) compared to their untreated control cells (12% and 2.5%) (*p* < 0.0001 and <0.001), respectively.

## 4. Conclusions

ZnO NPs are exceptional and increasingly applied in the study and therapy of cancer. ZnO NPs can be a successful alternative for conventional cancer treatment with selective targeting properties and utility as carrier agents. The analysis focused primarily on ZnO NPs, the relationship between ZnO NPs and triple-negative cancer cell lines, and the probable mechanism of ZnO NPs in human body biology, resulting in their localization and cytostatic of cancer cells. Although ZnO NPs cause cytotoxicity in cancer cells through the generation of oxidative stress, this may not be the primary cytotoxicity mechanism but rather a zinc-mediated protein response. In this study, the induction of intracellular ROS, which could specifically influence the mechanical process of cell viability, was seen to lead to substantial cytotoxicity of human ovarian cells by the induction of ZnO NPs 50–60 nm. The results on the cellular metabolic activities using the MTT assay showed a significant difference in the IC_50_ values between MCF-7/WT cells compared to MCF-7/DR by 1.85 ± 0.4 µg/mL to 1.5 ± 0.2 µg/mL, respectively. At the same time, the IC_50_ of the MDA-MB-231 cell line was two folds higher than that of the MCF-7 cell lines. Furthermore, the IC_50_ results did not show significant differences between MDA-MB-231/WT cells 4.2 ± 0.6 µg/mL compared to MDA-MB-231/DR cells 4.0 ± 1.1 µg/mL.

## Figures and Tables

**Figure 1 molecules-27-01827-f001:**
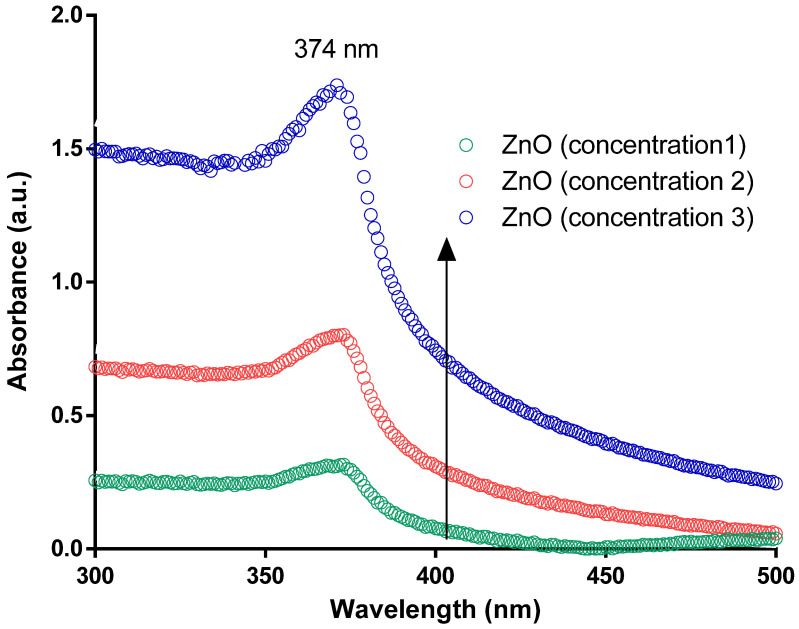
UV-vis spectrum of the generated and rigorously purified ZnO NPs *Citrullus colocynthis* (L.) Schrad leaves extract suspended in aqueous solution as measured at ambient temperature with a single peak corresponding to the generation of ZnO NPs with maximum absorbance at λ_374_ nm. The spectrum suggests a complete reduction of Zn cations using leaf extract, and the generated NPs were formed regardless of the concentration of plant extract added to the reaction mixture.

**Figure 2 molecules-27-01827-f002:**
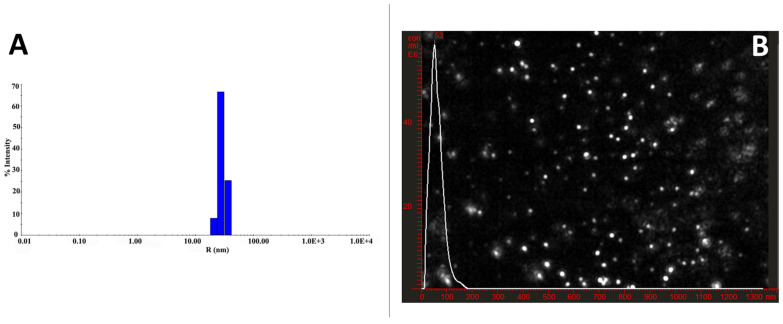
Characterization of ZnO NPs. (**A**) representative histogram of the size (diameter) distribution by the number of ZnO nanoparticle (**B**) A video frame of nanotracker analysis (NTA) with a single peak measurement and an average of 53 ± 3.5 nm without any visual aggregations.

**Figure 3 molecules-27-01827-f003:**
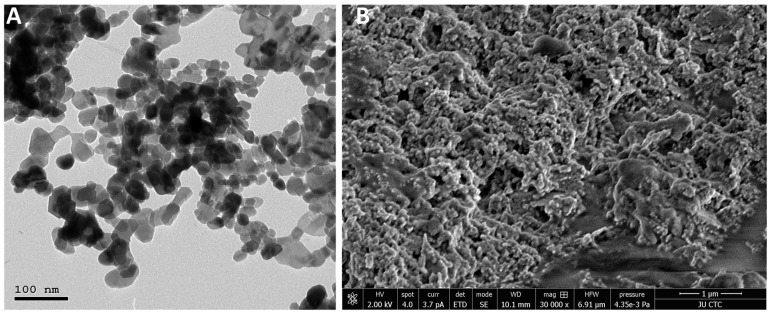
Representative TEM (**A**) and SEM (**B**) images indicate the formation of spherical nanoparticles using the green synthesis approach with a particle size between 50–60 ± 5 nm in diameter. SEM analysis further confirmed that the 3D arrangement of the generated particles was spherical with monodisperse particle distribution.

**Figure 4 molecules-27-01827-f004:**
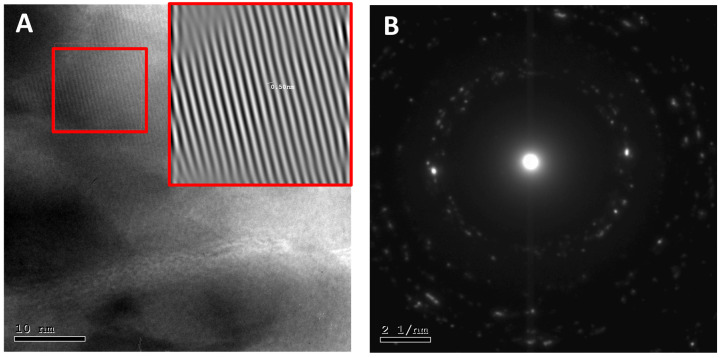
High-resolution TEM images (**A**) confirming the crystallinity of the generated particles using the green synthesis approach, (**B**) selected area diffraction (SAED) diffraction pattern of ZnO nanoparticles.

**Figure 5 molecules-27-01827-f005:**
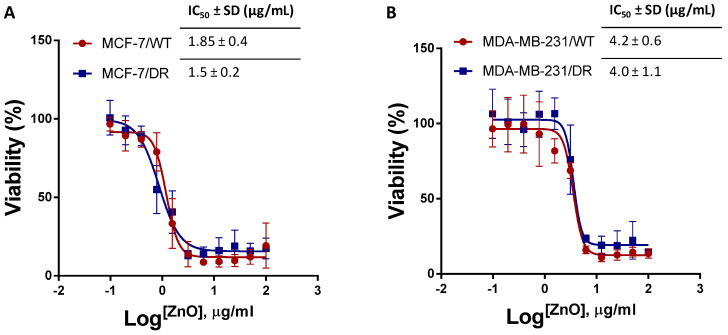
Dose–response curves. (**A**) Dose–response curve with IC50 of MCF-7/WT and MCF-7/DR cells after treatment with different concentrations of ZnO NPs for 72 h using the MTT assay. (**B**) Dose–response curve with IC50 of MDA-MB-231/WT and MDA-MB-231/DR cells after treatment with different concentrations of ZnO NPs for 72 h using the MTT assay (*n* = 3).

**Figure 6 molecules-27-01827-f006:**
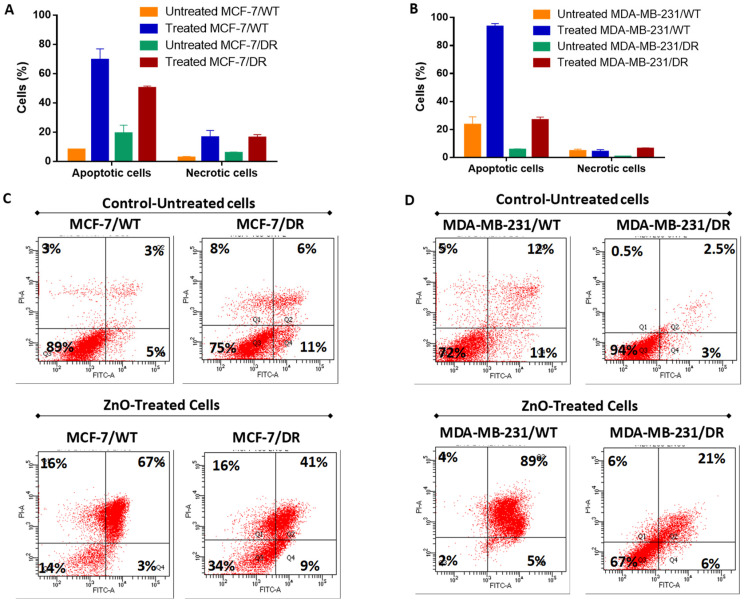
Flow cytometric analysis of the cell death modality. Breast cancer cell lines: MCF-7/WT MCF-7/DR, MDA-MB-231/WT, and MDA-MB-231/DR cells were treated 24 h with 10 µg/mL ZnO NPs and stained with FITC-conjugated annexin V and PI-stained, compared to the untreated control cells. (**A**,**B**). Average (%) of apoptotic and necrotic cells after treatment. (**C**,**D**) The quadrants in the dot plot indicate viable cells (lower left quadrant), early apoptotic cells (lower right quadrant), necrotic cells (upper left quadrant), and late apoptotic cells (upper right quadrant) (*n* = 3).

**Table 1 molecules-27-01827-t001:** Summarizes the hydrodynamic radius of the generated nanoparticles with their corresponding polydispersity index and the ζ-potential and particles size as obtained from the TEM images (*n* = 3).

DLS Size (d.nm)	The Polydispersity Index (PDI)	ζ-Potential (mV)	TEM Size (d.nm)
65 ± 5	0.3 ± 0.03	−6 ± 1	22 ± 3

## Data Availability

Not applicable.

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
