# Peer review of "Synthesis, Characterization, and Assessment of Anti-Cancer Potential of ZnO Nanoparticles in an In Vitro Model of Breast Cancer"

_molecules, 2022, doi:10.3390/molecules27061827_

Round 1
Reviewer 1 Report
The manuscript reported a one-step method using a greener, environmentally friendly, safer approach to generate monodisperse zinc oxide (ZnO) nanoparticles (NPs) using the leaf extract of Citrullus colocynthis (L.) Schrad. Subsequently, the ZnO nanoparticles were characterized by a series of methods and their antitumor activity was evaluated. However, there are similar strategies based on green methods approach in other references (DOI: 10.3390/molecules26133864, DOI: 10.2147/IJN.S186888, DOI: 10.1016/j.micpath.2019.04.022). The innovation of experimental design needs to be improved. Compared with the other reports, there were no obvious advantages in therapeutic effect. Hence, the novelty of the manuscript may not sufficient to publish on Molecules. Besides, there were some questions in the experiments and results parts of this manuscript.
- There are similar strategies and experimental designs in other reference Kavazet al synthesized ZnO NPs through a stable, simple, and eco-friendly green route via the use of A. lebbeck stem bark extract in 2018(DOI: 10.2147/IJN.S186888). No improvements and advantages were exhibited in this manuscript.
- In order to better understand the article, please add a chemical mechanismof synthesis of ZnO NPs by Citrullus colocynthis (L.).
- In Figure 2and Figure 3, the generated nanomaterials were characterized by nanotracker analysis (NTA), transmission electron microscopy (TEM) and so on. However, the experimental control group was missing. In order to ensure the accuracy of the novel strategy for synthesis of ZnO NPs reported in this manuscript, it is necessary to set up a control group using conventional synthesis methods.
- In the part of “1”, X-ray diffraction pattern and FTIR results should be supplemented in order to improve the characterization of ZnO NPs.
- Please update the references.
- In Figure 5, IC50sresults did not show significant difference between MDA-MB-231/WT cells compared to MDA-MB-231/DR However, the average (%) of apoptotic and necrotic cells after treatment was significantly different between the same two groups in Figure 6. Please explain these results or elucidate other modality of cell death.
Author Response
Reviewer 1:
The manuscript reported a one-step method using a greener, environmentally friendly, safer approach to generate monodisperse zinc oxide (ZnO) nanoparticles (NPs) using the leaf extract of Citrullus colocynthis (L.) Schrad. Subsequently, the ZnO nanoparticles were characterized by a series of methods and their antitumor activity was evaluated. However, there are similar strategies based on green methods approach in other references (DOI: 10.3390/molecules26133864, DOI: 10.2147/IJN.S186888, DOI: 10.1016/j.micpath.2019.04.022). The innovation of experimental design needs to be improved. Compared with the other reports, there were no obvious advantages in therapeutic effect. Hence, the novelty of the manuscript may not sufficient to publish on Molecules. Besides, there were some questions in the experiments and results parts of this manuscript.
Response:
We strongly believe that our current manuscript entitled “Synthesis, Characterization, and assessment of the anti-cancer potential of ZnO Nanoparticles in an in-vitro model of Breast Cancer” reports significant and novel findings which will attract a high level of readership and high volume of citations. However, the references that have been highlighted to us report the effect of ZnO nanoparticles on cancer cells. However, our particles were tested on drug resistance cells, and this is a new approach and added novelty to our study. Other studies were in just cancer cell lines.
Also, there is a need to utilize the ZnO-based system to address the invasive triple-negative breast cancer and address the aggressive phenotype characterized by a significantly high level of metastasis, a poor prognosis, and unpredictability in therapy due to conventional chemotherapy suppressing malignant cells is the main rationale for this work. Publication of this work will trigger futher research in this area and human clinical trials.
- There are similar strategies and experimental designs in other reference Kavazet al synthesized ZnO NPs through a stable, simple, and eco-friendly green route via the use of A. lebbeck stem bark extract in 2018(DOI: 10.2147/IJN.S186888). No improvements and advantages were exhibited in this manuscript.
Response:
We strongly believe that our current manuscript entitled “Synthesis, Characterization and assessment of the anti-cancer potential of ZnO Nanoparticles in an in-vitro model of Breast Cancer” reports significant and novel findings which will attract high level of readership and high volume of citations. Although the references highlighted to us, report the effect of ZnO nanoparticles on cancer cells. However, our particles were tested on resistance cells, and this is a totally new approach and can’t be considered just cancer cell lines.
- In order to better understand the article, please add a chemical mechanismof synthesis of ZnO NPs by Citrullus colocynthis (L.).
Response:
The following statement has been added to the manuscript to highlight
“The established hypothesis for the formation of NPs in this technique is a phytochemical-driven response wherein the leaf extracts comprise diverse reducing chemicals including enzymes, antioxidants, and phenolic moieties that convert zinc cations into ZnO NPs. The presence of phytochemicals promotes the postulated reduction of Zn(NO3)2.6H2O to generate zerovalent zinc, which further leads to the agglomeration of zinc atoms to nanosized particles that are ultimately stabilized by phytochemicals to produce isotropic (spherical) ZnO.”
- In Figure 2and Figure 3, the generated nanomaterials were characterized by nanotracker analysis (NTA), transmission electron microscopy (TEM) and so on. However, the experimental control group was missing. In order to ensure the accuracy of the novel strategy for synthesis of ZnO NPs reported in this manuscript, it is necessary to set up a control group
using conventional synthesis methods.
Response:
We believe setting uncomparable nanoparticles synthesiszed using the chemical approach as a control. The techniques and the analysis was generated from the particles we produced and also the one we used to study
- In the part of “1”, X-ray diffraction pattern and FTIR results should be supplemented in order to improve the characterization of ZnO NPs.
Response:
In general terms, SAED and XRD are the same. Both differ on the wavelength of the diffracted beam and are governed by the Bragg equation. We have the SAED pattern presented in Figure 4 panel B, but we do not have access to FTIR, and we believe it would not affect our findings and report
- Please update the references.
Response:
All references have been updated
- In Figure 5, IC50sresults did not show a significant difference between MDA-MB-231/WT cells compared to MDA-MB-231/DR However, the average (%) of apoptotic and necrotic cells after treatment was significantly different between the same two groups in Figure 6. Please explain these results or elucidate other modalities of cell death.
Our response:
We thank the respected reviewer for drawing our attention to this point. Indeed, the IC50 values were close for the MDA-MB-231/WT compared to MDA-MB-231/DR cells, while the % of apoptotic and necrotic cells were lower in MDA-MB-231/DR compared to MDA-MB-231/WT cells. In the MTT assay, we treated both cell lines with different concentrations of ZnO NPs for 72 hours, while in the apoptosis assay, we treated both cell lines with 10 µg/ml for 24 hours. Therefore, different incubation times and treatment concentrations can be possible reasons for such variation. Moreover, previously, we showed different mechanisms of doxorubicin resistance in MCF-7 compared to MDA-MB-231 cells (Alkaraki et al., 2020, Alshaer et al., 2019) may induce different responses in MDA-MB-231/DR cells compared to MCF-7/WT, MCF-7/DR, and MDA-MB-231/WT. We agree that a future study that deeply investigates molecular mechanistic and cell death modality induced ZnO NPs in drug-resistant cancer cells can be of high value and provide insights of developing potent therapeutics against drug-resistant tumors.
ALKARAKI, A., ALSHAER, W., WEHAIBI, S., GHARAIBEH, L., ABUARQOUB, D., ALQUDAH, D. A., AL-AZZAWI, H., ZUREIGAT, H., SOULEIMAN, M. & AWIDI, A. 2020. Enhancing chemosensitivity of wild-type and drug-resistant MDA-MB-231 triple-negative breast cancer cell line to doxorubicin by silencing of STAT 3, Notch-1, and beta-catenin genes. Breast Cancer, 27, 989-998.
ALSHAER, W., ALQUDAH, D. A., WEHAIBI, S., ABUARQOUB, D., ZIHLIF, M., HATMAL, M. M. & AWIDI, A. 2019. Downregulation of STAT3, beta-Catenin, and Notch-1 by Single and Combinations of siRNA Treatment Enhance Chemosensitivity of Wild Type and Doxorubicin Resistant MCF7 Breast Cancer Cells to Doxorubicin. Int J Mol Sci, 20.
Reviewer 2 Report
Manuscript by Alaa A. A. Aljabali et al, on “Synthesis, Characterization, and assessment of anti-cancer potential of ZnO Nanoparticles in an in-vitro model of Breast Cancer” describes the synthesis, and characterization of ZnO NPs for their cytotoxic effect on MCF-7 and MDA-MB-231 breast cancer cell lines.
The authors used fresh leaves from the Wadi Araba -south of Jordan were collected from Citrullus colocynthis (L.) Schrad, better to add where the specimen was deposited (national or international Herbarium) and if any acknowledgment/accession number.
They used extracts of leaves of Citrullus colocynthis (L.) Schrad, any details on the chemistry of compounds as active principles of plants.
Anticancer results on two cell lines can be presented in the form of a table.
Author Response
Reviewer 2
Manuscript by Alaa A. A. Aljabali et al, on “Synthesis, Characterization, and assessment of anti-cancer potential of ZnO Nanoparticles in an in-vitro model of Breast Cancer” describes the synthesis, and characterization of ZnO NPs for their cytotoxic effect on MCF-7 and MDA-MB-231 breast cancer cell lines.
The authors used fresh leaves from the Wadi Araba -south of Jordan were collected from Citrullus colocynthis (L.) Schrad, better to add where the specimen was deposited (national or international Herbarium) and if any acknowledgment/accession number.
They used extracts of leaves of Citrullus colocynthis (L.) Schrad, any details on the chemistry of compounds as active principles of plants.
The hypothetical mechanism have been added to the manuscript as follow
Response:
The following statement have been added to the manuscript to highlight
“The established hypothesis for the formation of NPs in this technique is a phytochemical-driven response wherein the leaf extracts comprise diverse reducing chemicals including enzymes, antioxidants, and phenolic moieties that convert zinc cations into ZnO NPs. The presence of phytochemicals promotes the postulated reduction of Zn(NO3)2.6H2O to generate zerovalent zinc, which further leads to the agglomeration of zinc atoms to nanosized particles that are ultimately stabilized by phytochemicals to produce isotropic (spherical) ZnO.”
Anticancer results on two cell lines can be presented in the form of a table.
Response:
Our response: we thank the respected reviewer for the kind advice. Indeed, for both breast cancer cell lines, the results of IC50 values can be expressed in a separate table. However, we think that expressing the results as appeared in figure 5 can provide a direct expression alongside with the figure thereby simplify understanding and analysis of results by reader.
Reviewer 3 Report
The manuscript reports interesting results concerning the green synthesis of ZnO nanoparticles their characterization and utilization their potential anti-cancer activity. Green synthesis is a simple, economic and safe way for the synthesis of Au NPs and considered as an effective alternative to the other physical and chemical approaches. This method have received increases attention in the last decades due to the potential biomedical application of green metal oxides nanoparticles. The manuscript is well written, well organized and the research work is supported with scientific references. However, I suggest certain suggestions and minor revision for quality enhancement, which are given bellow.
- The manuscript contains grammatical and typing mistakes which should be carefully read and remove and the language need improvement.
- Correct the numerical value in the introduction “3,37 eV” as “3.37 eV”. i.e insert dot instead of comma.
- The sentence in the introduction section “Sol-gel processing, homogeneous precipitation, mechanical milling, organometallic synthesis, the microwave approach, spray pyrolysis, thermal evaporation, and mechanochemical synthesis are all strategies for producing ZnO nanoparticle NPs”. The authors claim that these methods are reported for the preparation of ZnO nanoparticles, hence every preparation methods require separate reference.
- Discuss some biological applications of green synthesized ZnO nanoparticles in the introduction.
- Introduce the selected plant and discuss its reported medical applications.
- 1M sodium hydroxide solution (NaOH); 10g in 100 mL of double-distilled (ddH2O) water? Is this correct? 1M of NaOH in 100 mL can be prepared by dissolving 4g in 100 mL.
- The solution was centrifuged at 8000 g for 30 minutes. What does g means? Write rpm.
- There is no detail that how green ZnO nanoparticles were prepared. No conditions were mentioned. Discuss the method in detail.
- Normally gold is coated on surface for better morphological result. Why copper is used for coating?
- Figure 2A is has very bad resolution. Provide clear image.
- The PDI value in the text and table are not same. Why?
- The authors haven’t performed the XRD analysis and claimed the hexagonal wurtzite crystal structure. On what base the authors claim?
- Cite the following papers to enrich the manuscript with scientific proofs: J. Mex. Chem. Soc. 2021, 65(3), Arabian Journal of Chemistry (2019) 12, 908–931, New J. Chem., 2020, 44, 13330—13343.
Author Response
Reviewer 3
The manuscript reports interesting results concerning the green synthesis of ZnO nanoparticles their characterization and utilization their potential anti-cancer activity. Green synthesis is a simple, economic and safe way for the synthesis of Au NPs and considered as an effective alternative to the other physical and chemical approaches. This method have received increases attention in the last decades due to the potential biomedical application of green metal oxides nanoparticles. The manuscript is well written, well organized and the research work is supported with scientific references. However, I suggest certain suggestions and minor revision for quality enhancement, which are given bellow.
- The manuscript contains grammatical and typing mistakes which should be carefully read and remove and the language need improvement.
Response:
The entire manuscript has been revised for grammatical, typos, and language
- Correct the numerical value in the introduction “3,37 eV” as “3.37 eV”. i.e insert dot instead of comma.
Response:
Thank you, the correction has been added to the manuscript
- The sentence in the introduction section “Sol-gel processing, homogeneous precipitation, mechanical milling, organometallic synthesis, the microwave approach, spray pyrolysis, thermal evaporation, and mechanochemical synthesis are all strategies for producing ZnO nanoparticle NPs”. The authors claim that these methods are reported for the preparation of ZnO nanoparticles, hence every preparation methods require separate reference.
Response:
Appropriate references have been added to each statement from 17-23
- Discuss some biological applications of green synthesized ZnO nanoparticles in the introduction.
Response:
A paraghraph with potential applications of ZnO particles have been added to the manuscript
- Introduce the selected plant and discuss its reported medical applications.
Response:
The following statement have been added to the manuscript to highlight the medicinal value of the plant
“Citrullus colocynthis (L.) Schrad has been used as a medicinal plant for the treatment of leprosy, diabetes, cough, common cold, bronchitis, asthma, joint pain, jaundice, toothache, and gastrointestinal diseases, including constipation, indigestion, dysentery, colic pain, and many other microbial infections were all linked to the plant.”
- 1M sodium hydroxide solution (NaOH); 10g in 100 mL of double-distilled (ddH2O) water? Is this correct? 1M of NaOH in 100 mL can be prepared by dissolving 4g in 100 mL.
Response:
Thank you for pointing this out. The correct 4g in 100 ml have been corrected in the manuscript
- The solution was centrifuged at 8000 g for 30 minutes. What does g means? Write rpm.
Response:
G means G-Force (RCF). It is more accurate to write the centrifugation speed in G force rather than rpm because the rotor size might differ, and g force will be different while the revolutions per minute stay the same
- There is no detail that how green ZnO nanoparticles were prepared. No conditions were mentioned. Discuss the method in detail.
Response:
The approach is described in section 2.3
- Normally gold is coated on surface for better morphological result. Why copper is used for coating?
Response:
These are commercial TEM grids, and we regularly use them as they tend to be more stable without causing any split in the grids during imaging
- Figure 2A is has very bad resolution. Provide clear image.
Response:
Higher resolution images have been provided
- The PDI value in the text and table are not same. Why?
Response:
Thank you for pointing this out it was a mistake and now been corrected to 0.3±0.03 in both the text and the table
- The authors haven’t performed the XRD analysis and claimed the hexagonal wurtzite crystal structure. On what base the authors claim?
Response:
This was based on SAED (selected area diffraction), as shown in Figure 2 panel B
- Cite the following papers to enrich the manuscript with scientific proofs: J. Mex. Chem. Soc. 2021, 65(3), Arabian Journal of Chemistry (2019) 12, 908–931, New J. Chem., 2020, 44, 13330—13343.
Response:
Suggested references have been added to the manuscript
Round 2
Reviewer 1 Report
The authors have made great efforts to address our comments and all of our questions have been fully answered. Hence, we recommend this manuscript to be accepted by “Molecules”.
Reviewer 3 Report
The authors have sufficiently improved the manuscript. However the language still nedd to be polished. Figure 2A resolution need further improvement. Manuscript can be accepted after the suggested incorporation